# Comparison of the Immunogenicity of the LZ901 Vaccine and HZ/su Vaccine in a Mouse Model

**DOI:** 10.3390/vaccines12070775

**Published:** 2024-07-15

**Authors:** Yaru Quan, Chunxia Liu, Xu Lu, Xi Kong, Shuai Yang, Jian Kong, Wenyan Wan, Kaiqin Wang, Kangwei Xu, Ling Peng

**Affiliations:** 1NHC Key Laboratory of Research on Quality and Standardization of Biotech Products, NMPA Key Laboratory for Quality Research and Evaluation of Biological Products, National Institutes for Food and Drug Control, Beijing 102629, China; yaruquan@nifdc.org.cn (Y.Q.);; 2Beijing Luzhu Biotechnology Co., Ltd., Beijing 101100, China

**Keywords:** herpes zoster, LZ901, gE-specific IgG antibody, cell-mediated immunity

## Abstract

Herpes zoster (HZ) is an infectious disease caused by the reactivation of varicella zoster virus (VZV), with 68% of cases occurring in adults over 50 years of age. HZ/su (Shingrix^®^) was approved by the Food and Drug Administration in 2017 for the prevention of HZ in individuals ≥ 50 years of age and showed very good protection from HZ. However, due to the use of the adjuvant AS01_B_, adverse reactions caused by Shingrix are a concern. Aluminum hydroxide is the most commonly used adjuvant and is widely used in a variety of vaccines. We developed a recombinant zoster vaccine (code: LZ901) consisting of a tetramer of VZV glycoprotein E (gE) and a human Fc fusion protein expressed in CHO cells, an immune complex-like molecule that can be adsorbed with an aluminum hydroxide adjuvant. We compared the immunogenicity of LZ901 with that of HZ/su in BALB/c mice. The results showed that LZ901 induced levels of gE-specific IgG antibodies comparable to those induced by HZ/su, and the results of FAMA titers further demonstrated their similar neutralizing antibody abilities. Most importantly, LZ901 induced higher levels of cell-mediated immunity (CMI) (which plays a decisive role in the efficacy of zoster vaccines) than HZ/su in BALB/c mice. The numbers of cytokine-producing T cells in LZ901-vaccinated mice were significantly greater than those in v-vaccinated mice, and the proportions of CD4^+^ and CD8^+^ T cells producing at least two types of cytokines in LZ901-vaccinated mice were significantly greater than those in HZ/su-vaccinated mice.

## 1. Introduction

Varicella zoster virus (VZV) causes varicella during childhood and then remains latent throughout one’s lifetime. Herpes zoster (HZ), or shingles, is caused by the reactivation of VZV. VZV enters local lymph nodes through the respiratory mucosa epithelium and replicates during primary infection. Lymphocytes infected with VZV then enter the blood circulation through the lymphatic circulation, peripheral blood mononuclear cells (PBMCs) are infected, and the virus spreads to the skin via the bloodstream, which is clinically manifested as varicella. After varicella is cured, the virus remains latent in the craniocerebral ganglia, spinal dorsal root ganglia, autonomic neurons, or enteric neurons. When the resistance of a body is reduced, the latent virus is reactivated, replicates, and migrates to the skin along peripheral nerves, which is clinically manifested as HZ. Varicella vaccines depend on humoral immunity to prevent VZV infection and cell-mediated immunity (CMI), which plays a therapeutic role in the management or elimination of reactive VZV in infected cells and is decisive for vaccine efficacy. Approximately one in three people develop HZ during their lifetime [1]. Acute HZ is characterized by a dermatomal rash and is associated with pain. HZ may progress to postherpetic neuralgia (PHN), which can persist and be refractory for years. Other complications include meningitis, myelitis, and transient ischemic attack (TIA) or stroke. The risk of HZ is associated with decreased CMI with increasing age and/or immunocompromising conditions [2,3,4]. Elderly and immunosuppressed people have a greater risk of VZV infection. People over the age of 85 years have an even greater risk of getting shingles, and more than 30% of those people will develop PHN [5,6]. Elderly patients or immunocompromised patients can also have repeated attacks of HZ, and the spread of skin lesions, accompanied by bacterial infections or verrucous hyperplasia, can also lead to virus resistance. In severe cases, multiple organs, such as the lungs, organs of the gastrointestinal tract, and brain, may even be involved, and hepatitis, pancreatitis, pneumonia, myocarditis, esophagitis, or peptic ulcers may occur before the appearance of an HZ rash, which can easily result in misdiagnosis. Globally, the cumulative incidence of HZ ranges from 2.9 to 19.5 cases per 1000 people, with females having a greater risk of developing HZ. In the United States of America (USA), approximately 1 million cases of HZ were reported, causing approximately USD 1.3 billion in medical costs annually [7,8]. The number of annual cases of HZ in China is relatively limited, and there are geographical differences according to annual national reports [9]. HZ and its associated complications of PHN can impact patient quality of life and cause a substantial economic burden to the healthcare system.

The current treatment strategy for HZ includes antiviral medications such as acyclovir, valacyclovir and famciclovir and symptom-related treatments such as pain management. However, these antiviral medications are not suitable for elderly and immunosuppressed patients because most antiviral medications are hepatotoxic or nephrotoxic. Therefore, an effective vaccine could significantly reduce the medical costs of HZ and the development of PHN. There are two types of HZ vaccines, namely, a live attenuated VZV vaccine (Zostavax^TM^, Merck, Kenilworth, NJ, USA) and a recombinant adjuvanted VZV glycoprotein E (VZV gE) with the adjuvant AS01 (HZ/su Shingrix^®^, GlaxoSmithKline, London, UK), which were approved by the US Food and Drug Administration (FDA) for healthy elderly people and for select patients who are severely immunocompromised or patients with immune diseases aged ≥19 years [10,11,12]. However, Zostavax^TM^ has not been available globally since 2020, as patients who are immunosuppressed and immunocompromised are considered to have contraindications [13]. Therefore, a non-live, adjuvanted recombinant vaccine was selected to potentially increase the efficacy of HZ vaccination in the elderly group and allow the vaccination of immunocompromised people to decrease the incidence of HZ. LZ901 is a new recombinant VZV vaccine that consists of a purified tetramer of recombinant VZV gE and a human IgG_1_ Fc fusion protein expressed in CHO cells and adsorbed with aluminum hydroxide adjuvants. The LZ901 molecule comprises four VZV-gE-Fc fusion proteins, similar to an immune complex, with two Fc fragments and four VZV gEs.

VZV gE is most abundantly expressed on the surface of the virus and infected cells [14,15]. As one of the most abundant glycoproteins of VZV, the conserved glycoprotein E contains potential neutralization epitopes and T-cell epitopes and is essential for viral replication and transmission [16]. As a type I membrane protein, gE is an essential glycoprotein for the production of infectious VZV particles and is also the glycoprotein with the strongest antigenicity and the highest content on the viral envelope and the infected cell membrane. Moreover, gE is also widely present on the surface of VZV particles, on the cell membrane of host cells and in the cytoplasm of host cells and can induce cellular immunity and humoral immunity. Although the vaccine antigen of both LZ901 and HZ/su (Shingrix^®^) is VZV gE, the major difference is that LZ901 is a tetramer of the gE-Fc fusion protein, and each molecule is composed of four gEs and two Fcs. As reported previously, the HZ/su vaccine plays a therapeutic role in the elimination of reactivated VZV-infected cells, indicating that VZV-specific CMI is crucial in controlling the onset and severity of HZ and in reducing the risk of PHN [17]. Therefore, the aim of this study was to evaluate the immunogenicity of LZ901 in CMI and humoral immunity in BALB/c mice and compare it with that of HZ/su (Shingrix^®^).

## 2. Materials and Methods

### 2.1. Preparation of the LZ901 Vaccine

LZ901 is a tetramer of the recombinant VZV gE and the human IgG1 Fc fusion protein expressed in CHO cells. The fusion protein is formed by an extracellular domain of the recombinant glycoprotein gE of a live attenuated VZV strain (Oka) and an Fc fragment of human immunoglobulin. The LZ901 cells, which contained the VZV-gE-Fc fusion protein genes, were incubated in an automatic bioreactor, and the cell debris was removed via deep filters. Then, a multistep liquid chromatography column was applied to purify the tetrameric recombinant VZV gE-Fc protein (containing two Fcs). A low-pH process and a nanofilter were used to inactivate and remove any viruses from the LZ901 bulk solution. LZ901 was diluted to 200 μg/mL with Tris buffer and then mixed with an equal volume of aluminum hydroxide adjuvant to make a vaccine for use (50 μg/0.5 mL). HZ/su (Shingrix^®^, GlaxoSmithKline Biological SA, Batch No. K2424) consists of 50 μg of purified gE with AS01_B_, an adjuvant system containing 3-O-desacyl-4-monophosphoryl lipid A (MPL) (50 μg) and QS-21 (50 μg) within liposomes [18]. The placebo was saline.

### 2.2. Mouse Immunization

This study was approved by the Institutional Animal Care and Use Committee at the National Institutes for Food and Drug Control, China (approval number: 2023(B)068). Female BALB/c mice aged 5–7 weeks were randomly assigned to three groups of 20 mice each (Table 1), and the mice were immunized twice via back subcutaneous injection of 500 μL of vaccine on days 0 and 21. Blood was collected, and serum was isolated 21 and 35 days after the first immunization. The mouse spleen was removed at 35 days for flow cytometry analysis.

### 2.3. Measurement of gE-Specific Antibody Titers

The levels of gE-specific IgG antibodies in mouse serum were determined by ELISA [19]. gE protein (Sino Biological, Inc., Beijing, China) was diluted with coating buffer to a final concentration of 2 μg/mL and then used to coat 96-well microplates. The plates were refrigerated at 4 °C overnight. After washing 3 times with phosphate-buffered saline-Tween (PBS-T) solution, the plates were blocked with 200 μL/well solution (1% bovine serum albumin [BSA] in phosphate-buffered saline [PBS]) at 37 °C for 1 h. After removing the blocking solution, the plates were incubated in twofold-diluted mouse serum (100 μL/well) starting at 100-fold at 37 °C for 1 h. After washing 3 times, horseradish peroxidase (HRP)-labeled anti-mouse IgG antibody (Jackson ImmunoResearch Laboratory Inc., PA, USA, 1:10,000) was added to each well, and plates were incubated at 37 °C for 1 h. After washing 3 times, 3,3′,5,5′-tetramethylbenzidine (TMB substrate (BD Biosciences, San Diego, CA, USA)) was added to the plate. Five minutes later, the reaction was stopped, and the absorbance was measured at 450 nm with a microplate reader (Infinite M200 PRO, China Tecan laboratory equipment Co., Ltd., Shanghai, China). IgG titers were defined as end-point dilutions greater than 3 times the mean OD value of the negative control serum. The negative control serum was a 1:100 dilution of the serum mixture used in the placebo group.

### 2.4. Measurement of VZV Antibody Titers

Using FAMA method for VZV antibody test, MRC-5 cells were infected with VZV (M.O.I = 0.1) and daily observations of lesions were carried out. When the lesion reached 70–90%, infected cells were collected and washed twice with PBS. The infected cells were resuspended with PBS and cell density adjusted to 3 × 10^5^/mL, and then added to 12-well glass slides, 10 μL/well, and left at 37 °C for 30 min. The slide containing infected cells was fixed in cold acetone for 10 min and left to dry to obtain the VZV virus antigen slide. After inactivating the serum sample in a 56 °C water bath for 30 min, PBS was used as the diluent to dilute the serum twice in a series (1:4–1:2048), then added to the VZV virus antigen slide, 10 μL/well, and placed in a wet box at 37 °C for 30 min. It was then soaked and washed with PBS 3 times, 5 min per time, and air dried. Diluted FITC-labeled goat anti-mouse IgG in an appropriate proportion (1:200) was added to the VZV virus antigen slide, 10 μL/well, and placed in a wet box at 37 °C for 30 min. It was then soaked and washed with PBS 3 times, 5 min per time, and air dried. One drop of glycerol buffer was added to each well, covered with a coverslip, and then the results observed under a fluorescence microscope (XDY-1, Shanghai Yanfeng Precision Instrument Co., Ltd., Shanghai, China). The reciprocal of the maximum dilution of the serum sample is recorded as the VZV antibody titer of the serum sample. Individuals with bright yellow green fluorescent rings on the surface of infected cells and VZV antibody titers of 4 or higher are considered positive, otherwise, they are considered negative.

### 2.5. Flow Cytometry Analysis

The spleens of the mice were removed, and the splenocytes were extracted using mouse lymphocyte separation liquid (Dakewe Biotech Co., Ltd., Shenzhen, China). The splenocytes were counted, and their concentrations were adjusted to 4.0 × 10^6^ cells/mL with RPMI-1640 medium supplemented with 10% fetal bovine serum. Single-cell suspensions (500 µL) were seeded in duplicate in 24-well cell culture plates (2 × 10^6^ cells/well) (ExCell Bio, Suzhou, China).

The cells were stimulated with 200 μg/mL gE solution or RPMI-1640 medium supplemented with 10% fetal bovine serum at 37 °C for 20 h. After 20 h of incubation, 1 μL of protein transport inhibitor (Brefeldin A, BD Biosciences, San Jose, CA, USA) was added to each well, and the plates were incubated at 37 °C for 4 h. Thereafter, the cell pellets were resuspended in 50 μL of Mouse BD Fc Block (BD Biosciences, USA) to block nonspecific staining caused by the fluorescent antibody Fc receptor. Then, the cells were incubated with fluorochrome-conjugated antibodies (against CD3, CD4, CD8, and CD40L) for cell surface staining. After cell fixation and permeabilization, the cells were incubated with fluorochrome-conjugated antibodies (for IL-2, IL-4, and IFN-γ) for intracellular staining. Finally, the cells were gated (FSC/SSC), and 50,000 T cells were analyzed with a FACS Canto^TM^ flow cytometer (BD Biosciences, USA) and FlowJo X 10.0.7 R2 (Tree Star Inc., San Carlos, CA, USA).

### 2.6. Statistical Analysis

Statistical analyses were performed with GraphPad Prism 8.0.0 software (GraphPad Software Inc., La Jolla, CA, USA). The data are expressed as the mean ± standard deviation. Serum antibody titers were analyzed by one-way analysis of variance (ANOVA). A Chi-square test was performed to compare the frequencies of VZV gE-specific T cells among the groups. *p*-values < 0.05 were considered to indicate statistical significance.

## 3. Results

### 3.1. gE-Specific Humoral Immune Responses

LZ901 is a new vaccine and is in the development stage. To evaluate its immunogenicity, we subcutaneously administered the vaccine to the backs of mice. Mouse sera were isolated for 21 and 35 days after the first immunization, and gE protein-specific antibody titers were measured as a simple readout of the quality and magnitude of the humoral responses induced by vaccination (Figure 1 and Table 2). We found that the VZV gE-specific antibody titers in the LZ901 and HZ/su groups were significantly greater than those in the placebo group at the two time points (*p* < 0.0001). After the booster immunization on the 21st day, the gE-specific antibody titers in both the LZ901 and HZ/su groups increased significantly (*p* < 0.0001, Figure 1). However, no significant differences in gE-specific antibody titers were observed between the LZ901 and HZ/su groups at the two time points (*p* > 0.05, Table 2). The results indicated that the humoral immunity stimulated by LZ901 was comparable to that stimulated by HZ/su. The placebo group had no gE protein-specific antibodies.

### 3.2. VZV Antibody Evaluation

In 1974, Williams et al. first reported the detection of IgG antibodies against chickenpox virus using the FAMA test [20]. Subsequently, the FAMA test gradually became the preferred method for serological diagnosis of VZV due to its high sensitivity and specificity, known as the “gold standard” for detecting chickenpox IgG antibodies, which can be used to evaluate the body’s immune neutralization ability [21,22]. Therefore, we used the FAMA test to detect VZV antibodies in serum and evaluate the vaccine’s immune-neutralizing ability (Figure 2 and Table 3). The results showed that after the 35th day of immunization, the average FAMA titers of the LZ901 group and the HZ/su group were 832 and 568, respectively, with significant differences compared to the placebo group (*p* < 0.0001, Figure 2). However, no significant differences in average FAMA titers were observed between the LZ901 and HZ/su groups (*p* > 0.05, Table 3). The results indicated that the antibody ability stimulated by LZ901 was comparable to that stimulated by HZ/su. The placebo group had no antibody-neutralizing ability.

### 3.3. gE-Specific CMI Responses

CMI plays a decisive role in the efficacy of zoster vaccines [13,17]. To investigate the CMI responses induced by LZ901 and HZ/su, the gE protein was used to stimulate splenocytes from mice 35 days after the first immunization, and the average absolute numbers of cytokine-producing CD4^+^ and CD8^+^ T cells per 50,000 T cells after immunization were calculated. The results are summarized in Figure 3. In the LZ901 group, we observed a significant increase in the number of T cells with each of the four functions after stimulation with gE protein (IFN-γ^+^CD4^+^/CD8^+^, *p* < 0.0001; IL-2^+^CD4^+^/CD8^+^, *p* < 0.0001; IL-4+CD4^+^/CD8^+^, *p* < 0.0001; CD40L^+^CD4^+^/CD8^+^, *p* < 0.001). Compared with those in unstimulated samples, only IFN-γ^+^CD4^+^/CD8^+^, IL-4^+^CD8^+^, and CD40L^+^CD8^+^ T cells in the HZ/su group were significantly different. The placebo group had no CMI response.

CMI was further confirmed by evaluating the frequencies of cytokine-producing CD4^+^ or CD8^+^ T cells. Both the LZ901 and HZ/su groups exhibited strongly enhanced CD4^+^ and CD8^+^ T-cell responses (Table 4). The fold increase in cytokine levels was 1.92-fold (*p* < 0.0001) for IL-2^+^CD4^+^ T cells, 2.08-fold (*p* < 0.0001) for IL-4^+^CD4^+^ T cells and 1.49-fold (*p* < 0.0001) for CD40L^+^CD4^+^ T cells in the LZ901 group compared with the HZ/su group. Moreover, LZ901 induced greater frequencies of gE-specific CD8^+^ T cells expressing IFN-γ (1.17-fold, *p* = 0.0165), IL-2 (2.27-fold, *p* < 0.0001), IL-4 (1.88-fold, *p* < 0.0001) and CD40L (1.75-fold, *p* < 0.0001) following immunization than did HZ/su. More data are shown in Table 4.

### 3.4. Profile of VZV gE-Specific Cytokines Produced by T Cells

A mouse was considered to exhibit a strong cellular immune response when the increase multiple (IM) of frequencies of CD4^+^ or CD8^+^ cells was ≥1.5 (Table 4), the cytokine was activated and the data were considered positive. In the LZ901 group, VZV gE-specific CD4^+^ T cells predominantly produced IFN-γ (95%), followed by IL-2 (85%), CD40L (85%) and IL-4 (80%). VZV gE-specific CD8^+^ T cells predominantly produced CD40L (90%), followed by IFN-γ (85%) and IL-2 (85%), and relatively lower levels of IL-4 (65%). In contrast, the predominant cytokine produced by the CD4^+^ T cells was IFN-γ (65%), followed by IL-4 (65%), IL-2 (45%) and CD40L (45%). Moreover, the predominant cytokine produced by the CD8^+^ T cells was CD40L (85%), followed by IFN-γ (65%), IL-4 (50%) and IL-2 (45%) in the HZ/su group.

## 4. Discussion

HZ is a common and exceptionally painful condition caused by the reactivation of latent VZV, the most common PHN, which develops with age and substantially affects the quality of life among elderly people [23,24]. HZ/su (Shingrix^®^), approved for prevention of HZ, has a remarkably high protective efficacy and durable protection.

In this study, we aimed to elucidate the immunogenicity of LZ901, a novel VZV gE-based vaccine candidate, by comparing it with the established HZ/su in a mouse model. We evaluated humoral immune response by analyzing gE-specific IgG antibody titers and VZV antibody titers and evaluated cell-mediated immune (CMI) response using flow cytometry analysis. The results showed that LZ901 induced levels of humoral immune response comparable to those induced by HZ/su. In addition, our findings underscore the potential of LZ901 to induce a robust cytokine response in CD4^+^ and CD8^+^ T cells, such as Th1-type cytokines (IFN-γ, IL-2) and Th2-type cytokines (IL-4) indicative of a potent cell-mediated immune (CMI) reaction, with higher percentages of cytokine-producing T cells compared to HZ/su. Specifically, 60% of VZV gE-specific CD4^+^ and CD8^+^ T cells in the LZ901 group produced four types of cytokines, whereas only 15% of CD4^+^ and CD8^+^ T cells in the HZ/su group achieved the same. More detailed results are shown in Figure 4. The results indicated that there were significantly more cytokines produced by VZV gE-specific T cells in the LZ901 group than in the HZ/su group. This disparity in cytokine production highlights the enhanced CMI of LZ901, which could translate to a more effective immune response against VZV reactivation.

The superior CMI response observed with LZ901 may be attributed to its unique formulation. LZ901’s design leverages the sophisticated mechanisms of the human immune system for the processing of foreign antigens. The vaccine incorporates two Fc fragments that engage with Fcγ receptors (FcγR), which are abundantly present on the surfaces of antigen-presenting cells (APCs), including dendritic cells, macrophages, and monocytes. This interaction allows the varicella-zoster virus glycoprotein E (VZV gE) to be actively internalized into cells via FcγR-mediated endocytosis. Once inside, the gE is degraded by intracellular proteases into polypeptides, which are subsequently presented on the cell surface in conjunction with major histocompatibility complex class II (MHC-II) molecules. This process may also occur through antigen cross-presentation, effectively priming the immune system [25,26].

Moreover, the use of aluminum hydroxide in LZ901 is noteworthy. Unlike the AS01_B_ adjuvant system in HZ/su, which contains MPL (acts as a TLR4 agonist and stimulates the NF-κB pathway and cytokine production and directly activates antigen-presenting cells) and QS-21 (natural saponin that can stimulate CMI and increase antibody production) [27,28,29,30,31,32,33,34,35], aluminum hydroxide is a more traditional and cost-effective adjuvant. The tetramer structure of VZV gE in LZ901, along with the aluminum hydroxide adjuvant, could facilitate more efficient antigen presentation and processing by antigen-presenting cells (APCs). This, in turn, may lead to a heightened activation of T cells and a more pronounced Th1 response, which is crucial for combating intracellular pathogens like VZV. The particle size and surface charge of aluminum hydroxide may also contribute to its adjuvanticity, although further research is needed to fully elucidate these mechanisms.

The VZV-specific T-cell response is the only key to controlling latent reactivation; therefore, attention must be paid to this for the development of a herpes zoster vaccine. The ability of LZ901 to induce a strong CMI, along with a humoral response comparable to HZ/su, positions it as a promising candidate for preventing HZ and its complications, including postherpetic neuralgia (PHN). The ongoing multicenter, randomized, double-blind, placebo-controlled phase III clinical trial in China and randomized, double-blind, placebo-controlled phase I clinical trial in the USA will provide valuable data on the safety, tolerability, and efficacy of LZ901 in humans.

## 5. Conclusions

In summary, our preclinical study suggests that LZ901 has the potential to offer a more effective immune response against VZV compared to the currently available HZ/su. The enhanced CMI response, coupled with a comparable humoral response and the use of a traditional adjuvant, makes LZ901 a strong contender in the field of VZV vaccination. The ongoing clinical trials will be instrumental in validating these findings and assessing the real-world impact of LZ901 on HZ prevention.

## Figures and Tables

**Figure 1 vaccines-12-00775-f001:**
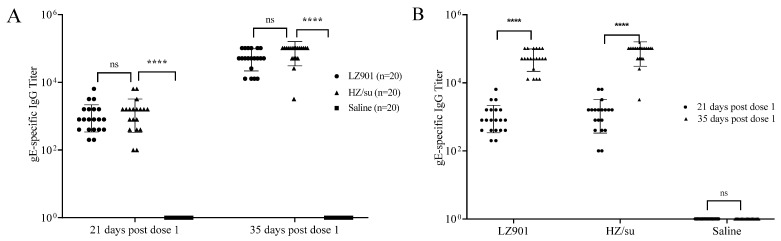
An analysis of gE-specific IgG against vaccines was performed on serum at 21 and 35 days after the first immunization, and gE-specific IgG titers against HZ/su, LZ901, and saline were measured: gE-specific IgG Titer by group (**A**) and gE-specific IgG Titer by time (**B**). The data points represent the values measured for each mouse, and the error bars indicate the average ± SD. Statistical analyses were conducted using ordinary one-way ANOVA. ns, *p* > 0.05; ****, *p* < 0.0001.

**Figure 2 vaccines-12-00775-f002:**
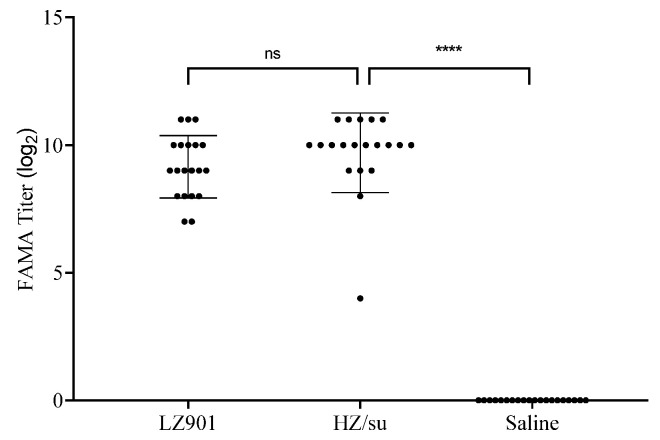
VZV antibody evaluation. Determination of VZV antibody titers in LZ901 vaccine group and HZ/su vaccine group using FAMA. The reciprocal of the maximum dilution of the serum sample is recorded as the VZV antibody titer of the serum sample. Individuals with bright yellow green fluorescent rings on the surface of infected cells and VZV antibody titers of 4 or higher are considered positive, otherwise, they are considered negative. The data points represent the values measured for each mouse, and the error bars indicate the average ± SD. Statistical analyses were conducted using ordinary one-way ANOVA. ns, *p* > 0.05; ****, *p* < 0.0001.

**Figure 3 vaccines-12-00775-f003:**
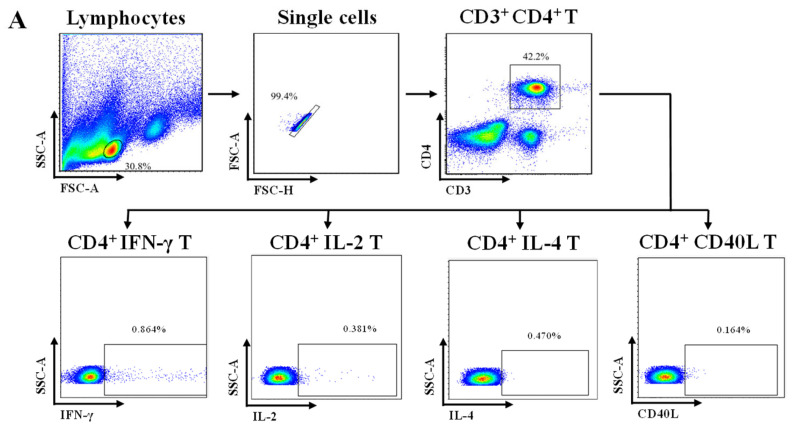
The number of CD4^+^ or CD8^+^ T cells expressing IFN-γ, IL-2, IL-4 and CD40L cytokines per 50,000 T cells was analyzed by flow cytometry. (**A**,**B**): Flow cytometry profiles showing selected windows and gating strategies applied to identify various cytokines in CD4^+^ T cells (**A**) and CD8^+^ T cells (**B**). (**C**,**D**): Number of CD4^+^ or CD8^+^ T cells expressing IFN-γ, IL-2, IL-4 and CD40L cytokines per 50,000 T cells analyzed by flow cytometry. VZV gE represents cells stimulated with the gE protein, and Unstimulated represents cells unstimulated with the gE protein. The error bars indicate the average ± SD. Statistical analyses were conducted using Fisher’s exact probability test. ns, *p* > 0.05; *, *p* < 0.05; **, *p* < 0.01; ***, *p* < 0.001; ****, *p* < 0.0001.

**Figure 4 vaccines-12-00775-f004:**
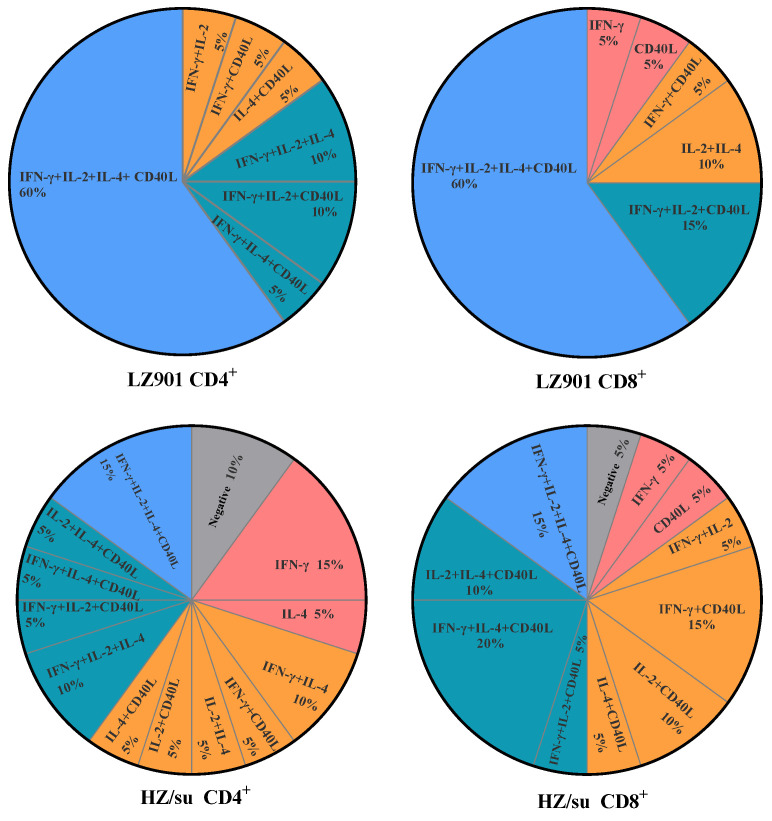
Distribution of cytokines in CD4^+^ and CD8^+^ T cells in the vaccine groups.

**Table 1 vaccines-12-00775-t001:** Vaccine and study groups.

Vaccine	Injected Protein Content	Number of Mice	Dilution Solution
LZ901	4.5 μg	20	Saline
HZ/su	4.5 μg	20	Saline
Saline	0 μg	20	/

**Table 2 vaccines-12-00775-t002:** Changes in gE-specific antibody titers and associated parameters during the immunization course.

Time Point	Median (Q1, Q3) of Antibody Titers		Average GMT		*p*
	LZ901	HZ/su	LZ901	HZ/su	
Baseline *	1	1	1	1	
Day 21	800 (400, 1600)	1600 (400, 1600)	857	1033	0.9128
Day 35	51,200 (32,000, 102,400)	102,400 (51,200, 102,400)	46,144	69,941	0.0829

* The female BALB/c mice used in the experiment were newborn mice with almost no antibodies in their bodies, so an analysis of gE-specific IgG against vaccines was not performed on serum at 0 days after the first immunization, which was directly defined as the baseline 1. Comparison of antibody titers between the LZ901 and HZ/su groups. Q1 and Q3 indicate the 25th and 75th percentiles, respectively.

**Table 3 vaccines-12-00775-t003:** Changes in FAMA titers and associated parameters during the immunization course.

Time Point	Median (Q1, Q3) of FAMA Titers		Average GMT		*p*
	LZ901	HZ/su	LZ901	HZ/su	
Baseline *	1	1	1	1	
Day 35	781 (128, 2048)	1114 (16, 2048)	568	832	0.2454

* The female BALB/c mice used in the experiment were newborn mice with almost no antibodies in their bodies, so an analysis of gE-specific IgG against vaccines was not performed on serum at 0 days after the first immunization, which was directly defined as the baseline 1. Comparison of FAMA titers between the LZ901 and HZ/su groups. Q1 and Q3 indicate the 25th and 75th percentiles, respectively.

**Table 4 vaccines-12-00775-t004:** Increase multiple (IM) of frequencies * for cytokine-secreting CD4^+^ and CD8^+^ T cells.

Items	Group	IM of Frequency for Cytokine-Secreting CD4^+^ T Cells				IM of Frequency for Cytokine-Secreting CD8^+^ T Cells			
		IFN-γ^+^ CD4^+^	IL-2^+^ CD4^+^	IL-4^+^ CD4^+^	CD40L^+^ CD4^+^	IFN-γ^+^ CD8^+^	IL-2^+^ CD8^+^	IL-4^+^ CD8^+^	CD40L^+^ CD8^+^
Range	LZ901	1.33–292.99	0.55–403.31	1.07–165.57	0.88–86.18	0.34–85.82	0.77–198.62	0.24–297.98	1.07–107.71
	HZ/su	0.1–68.52	0.42–75.69	0.17–72.28	0.27–51.26	0.05–56.73	0.03–49.97	0.02–117.28	0.1–56.10
	Saline	0.1–1.21	0.04–1.06	0.21–1.06	0.09–1.06	0.05–1.32	0.02–1.32	0.01–1.32	0.05–1.32
Mean	LZ901	36.37	61.92	52.82	20.58	29.99	40.83	65.99	29.84
	HZ/su	10.43	8.72	12.32	5.97	9.58	12.17	22.96	8.45
	Saline	0.9	0.93	0.84	0.68	1	0.85	0.77	0.76
SD	LZ901	67.13	88.58	55.92	26	24.02	58.17	97.73	38.01
	HZ/su	21.31	18.33	23.14	12.42	16.16	17.68	38.25	13.02
	Saline	0.28	0.22	0.27	0.38	0.27	0.41	0.43	0.42
Median	LZ901	7.93	46.67	22.81	6.17	30.47	14.79	11.61	3.81
	HZ/su	2.17	1.415	1.845	1.195	1.195	1.095	1.615	0
	Saline	0.98	0.99	0.98	0.97	1.04	1.03	0.99	0.99
Positive rate ^#^	LZ901	19/20 (95%)	17/20 (85%)	16/20 (80%)	17/20 (85%)	17/20 (85%)	17/20 (85%)	13/20 (70%)	18/20 (85%)
	HZ/su	13/20 (70%)	9/20 (45%)	12/20 (60%)	9/20 (45%)	13/20 (65%)	9/20 (45%)	10/20 (50%)	17/20 (85%)
	Saline	0	0	0	0	0	0	0	0

* The calculation process of IM: the control group without stimulation of each type of CD4^+^ and CD8^+^ cell was calculated as the background. The number of CD4^+^ or CD8^+^ cells in each vaccine group was divided by the total number of CD4^+^ or CD8^+^ T cells as the proportion of CD4^+^ and CD8^+^ cells. The frequency of CD4^+^ or CD8^+^ cells (Freq) was calculated by subtracting the corresponding background from the proportion of CD4^+^ or CD8^+^ cells. If the proportion of CD4^+^ or CD8^+^ cells in the vaccine group was less than or equal to the background value, it was imputed to one CD4^+^ or CD8^+^ cell per corresponding CD4^+^ or CD8^+^ T cell. Finally, the frequency of CD4^+^ or CD8^+^ cells after stimulation was divided by the frequency of CD4^+^ or CD8^+^ cells without stimulation to calculate the IM of CD4^+^ or CD8^+^ cells. ^#^ A mouse was considered to exhibit a strong cellular immune response when the IM of CD4^+^ or CD8^+^ cells was ≥1.5, the cytokine was activated, and the data were considered positive. The positive data for the activation markers of the mice in the vaccine group were recorded as 1, the negative data were recorded as 0, and the data were used to count the activation markers of each mouse.

## Data Availability

All the data, materials, and methods used in the analysis are available from the corresponding author upon request.

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
