# Peer review of "Comparison of the Immunogenicity of the LZ901 Vaccine and HZ/su Vaccine in a Mouse Model"

_vaccines, 2024, doi:10.3390/vaccines12070775_

Round 1

Reviewer 1 Report (Previous Reviewer 3)

Comments and Suggestions for Authors

1. Since the authors mentioned that one of concerns caused by Shingrix is the adjuvant AS01B, and the authors have applied Aluminum hydroxide. Have the authors tried to compare the Aluminum hydroxide with AS01B  to rule out that Aluminum hydroxide will not bring similar adverse reactions in presence of HZ/su core immunized part?

2. Why do authors choose 21 days and 35 days to collect mouse sera after the first immunization?

Have you ever tried to collect samples at longer days such as 50 days and 70 days?

3. In the Table 4, I am wondering why the ranges of the LZ901 and HZ/su are so large? For example, the range of IFNgamma CD4 treated with LZ901 is from 1.33-292.99. Do authors have some explanations for them? In addition, since the range is so large, does the mean have the biological meaning?

Author Response

Reviewer 2 Report (Previous Reviewer 2)

Comments and Suggestions for Authors

The authors have used FAMA test, which they suggest detect neutralizing antibodies. The way test is described does not appear to detect neutralizing antibodies. In fact, it is presumably detecting anti VZV antibodies (which may or may not be neutralizing).

Authors need to modify the text to reflect this.

Neutralization assay for VZV is difficult to do to to cell association of virus and due to low growth of virus in cell culture. Since detection of VZV antibodies has been used as a alternate measure, the results presented appear meet the requirement

Author Response

Reviewer 3 Report (Previous Reviewer 1)

Comments and Suggestions for Authors

This manuscript describes an alternative varicella zoster vaccine that avoids the adjuvant AS01B and uses aluminum hydroxide in adjuvant LZ901. A tetramer of glycoprotein E was the basis of its activity, and this tetramer was attached to a human Fc fusion protein. 

Comments on the revised manuscript:

2.4, lines 152-3: please define “appropriate proportion” 

3.2, line 223, 35st should be 35th

Author Response

This manuscript is a resubmission of an earlier submission. The following is a list of the peer review reports and author responses from that submission.

Round 1

Reviewer 1 Report

Comments and Suggestions for Authors

This manuscript describes a recombinant herpes zoster (shingles) vaccine (LZ901) from a cleverly designed tetramer of the VZV glycoprotein E (gE) and a human fusion protein, Fc from IgG1. The resulting protein was expressed in CHO cells and resembles an immunoprotein. This was developed with the common adjuvant aluminum hydroxide in response to concerns about the adjuvant AS01B used with the commercial herpes zoster vaccine, Shingrix®.  

There is a typo in the abstract that calls gE a VSV protein instead of a VZV protein. 

The new vaccine LZ901 showed advantages over Shingrix® in BALB/C mice in several specific ways, including higher levels of cell-mediated immunity through greater cytokine-producing T-cell production and proportions of CD4+ and CD8+ T cells that produced at least two types of cytokines being significantly higher than with Shingrix®.  

Materials and methods 

The correct abbreviation for milliliters is mL, not ml. There should be a space between the number and the °C symbol for temperature.  

Results 

Figure 2. I do not understand why the minima to maxima are indicated for the data, but the standard deviations (or variances) are not shown. Providing one or the other statistical measure would make the probability test data easier to understand. I understand that the information is available in Table 3, but Figure 2 is not much use without it or without a comment saying, “See Table 3 for data standard deviations and means.” 

Discussion 

Font size changes on lines 279-280. 

Overall 

This manuscript is well-written and easy to understand. The data are presented effectively (see my minor comments above), and the English is excellent. The ultimate result of this study is that a promising new vaccine candidate is undergoing clinical trials in China and the USA. The more immediate result is enhanced cellular immunity with the promise of decreased clinical issues compared with Shingrix®. I recommend publication with minor revisions.

Reviewer 2 Report

Comments and Suggestions for Authors

The authors have produced tetrameric (fused to human IgG1Fc) glycoprotein E of VZV in CHO cells and used along with aluminum hydroxide adjuvant as vaccine (LZ901) against VZV. Compared to existing recombinant vaccine, (Shingrix), LZ901 induced higher levels of cell mediated immune response in mice.

The  idea is neither novel nor original.  A number of studies have used recombinant antigen (gB, gE) vaccine to induce VZV specific both humoral/cellular immune responses (Vaccines 202412(3), 333). The authors have used tetrameric form of gE  (do not explain why such form woulds be more immunogenic) , when in virus it forms heterodimer with glycoprotein gI. Nevertheless, testing of different forms and combinations of vaccine antigen is required to get the best vaccine candidate

Concern

Since both humoral & cellular immune responses are required to prevent VZV infection, authors should provide data regarding induction of neutralizing antibody responses

Minor

line 18----VSV should be VZV

Reviewer 3 Report

Comments and Suggestions for Authors

The authors aim to improve the treatment to herpes zoster (HZ) since the current patient number in the United States is 1 million and 1.3 billion dollars was spent to treat these diseases annually but lack of highly efficient treatment that can be applied to the elderly healthy people and patients who suffer the immunocompromised diseases. The authors compared an improved version of HZ/su, named LZ901, to HZ/su, using VZV gE stimulated T cells producing specific cytokines. The results showed that a sharply increase at the cell number of LZ901 immunizations’ mice in secreting IL-4, IL-2, and CD40L. In addition, the LZ901 vaccinated mice showed more statistically significant in IFN-γ positive cells (four asterisks) compared to the HZ/su vaccinated mice (two asterisks). The work is promising but some concerns and questions are shown below:

1. In Figure1A, can the authors add the number of mice that is behind each treatment using parathesis? For example, LA901 (n = X), which will make the readers easier to read and compare.

2. In Figures 2C and 2D, upper two panels (IFN-γ positive cells/50,000 T cells and IL-2 positive cells/50,000 T cells), can the author explain why there is a very large variation in saline treatment, the error bars are very large since they are placebo control groups?

3. In the discussion part, can the authors clarify further that “ the processed antigen is presented to CD4+ or CD8+ T cells to induce an immune response dominated by Th1/typ2 T helper (Th2) cells.” Since the traditional CD4 T cells comprise of Th1 helper cells, Th2 helper cells, how the immune response here can be dominated by Th1/Th2 cells?

4. In the paper, the authors use the CD4+ T cells. Since the Treg cells are also CD4 positive, I am wondering if the authors can distinguish the traditional CD4 or regulatory T cells since the Treg cells are also CD4 positive.
